# Évaluation de méthodes d'attribution de mots-clés d'astronomie standardisés à des résumés d'articles d'héliophysique

Liza Fretel[1]   Corentin Louis[1]   Baptiste Cecconi[1]

(1) LIRA, Observatoire de Paris, Université PSL, Sorbonne Université, Université Paris Cité, CY Cergy Paris Université, CNRS, 92190 Meudon, France

`liza.fretel@obspm.fr`, `corentin.louis@obspm.fr`, `baptiste.cecconi@obspm.fr`

## RÉSUMÉ

Cette étude s'intéresse à l'attribution de mots-clés issus des concepts de l'Unified Astronomy Thesaurus (UAT) à partir de titres et résumés d'articles dans le domaine astrophysique. Elle soulève des problématiques de classification multi-labels extrême car les labels sont très épars (2411 labels possibles pour moins de dix labels positifs) et d'un manque de données d'entraînement qualitatives. Plusieurs méthodologies ont été évaluées : application du modèle KAILAS sur nos données ; entraînement d'un vectoriseur TF-IDF suivi d'une régression linéaire ; vectorisation des champs textuels des concepts UAT avec AstroBERT ; entraînement d'une architecture R-GCN ; correspondance de chaîne de caractères. Pour ces expériences, nous avons collecté un corpus de 34 025 résumés d'articles d'astrophysique. 5 361 de ces articles contenaient au moins un mot-clé sous le concept d'héliophysique. Nous avons aussi utilisé un petit corpus (31 documents) de résumés d'articles prépubliés dans notre sous-domaine d'intérêt (l'héliophysique) qui ont manuellement été annotés avec des mots-clés. Sur le corpus ADS héliophysique, KAILAS a obtenu le meilleur score F1 atteignant 0.5453 (contre 0.5006 pour le TF-IDF avec régression linéaire), et le TF-IDF avec régression linéaire a obtenu un score F1 de 0.6612 sur les prépublications d'héliophysique, contre 0.3612 pour KAILAS.

## ABSTRACT

**Evaluation of Methods for Assigning Standardized Astronomy Keywords to Heliophysics Papers' Abstracts**

This study focuses on assigning keywords from the Unified Astronomy Thesaurus (UAT) concepts to titles and abstracts of articles in the field of astrophysics. It raises issues of extreme multi-label classification due to the highly dispersed nature of the labels (2411 possible labels with less than ten positive labels) and a lack of qualitative training data. Several methodologies were evaluated : testing the KAILAS model on our data ; training a TF-IDF vectorizer followed by linear regression ; vectorization of the text fields of the UAT concepts using AstroBERT ; training an R-GCN architecture ; and string matching. For these experiments, we collected a corpus of 34,025 abstracts of astrophysics articles. 5,361 of these articles contained at least one keyword under the concept of heliophysics. We also used a small corpus (31 documents) of abstracts of pre-published articles in our subfield of interest (heliophysics) that were manually annotated with keywords. On the ADS heliophysics corpus, KAILAS obtained the best F1 score, reaching 0.5453 (versus 0.5006 for the TF-IDF with linear regression), and the TF-IDF with linear regression obtained a F1 score of 0.6612 on pre-published heliophysics articles, versus 0.3612 for KAILAS.

MOTS-CLÉS : classification multi-label extrême, suggestion de mots-clés, astrophysique.

KEYWORDS: extreme multi-label classification, keywords suggestion, astrophysics.

# 1  Introduction

## 1.1  Contexte

L'Unified Astronomy Thesaurus (Accomazzi et al., 2014), ou UAT, est un artéfact sémantique mettant à disposition des concepts de l'astronomie organisés en hiérarchie. À la version 6.0.0 du thésaurus, ces concepts sont répartis en 11 sous-domaines de l'astronomie, incluant par exemple : "Astrophysical processes", "Cosmology", "Exoplanet astronomy", "Galactic and extragalactic astronomy", "Heliophysics". Chaque concept de l'arbre est accompagné d'un label et parfois d'une définition. L'UAT est entretenu par la communauté astronomie, qui organise des discussions pendant lesquelles de nouvelles définitions, de nouveaux concepts ou des modifications structurelles sont proposées. Depuis plusieurs années déjà, il est recommandé aux auteurs et éditeurs d'articles d'astrophysique d'utiliser des mots-clés standardisés issus de l'UAT (Chen et al., 2021) afin d'en faciliter leur découverte. Par exemple, depuis 2019, la revue AAS (American Astronomy Society) classifie toutes ses publications à l'aide de mots-clés UAT [1].

Se posent alors deux questions : comment annoter les articles retrospectivement, et comment aider les futurs auteurs et éditeurs à sélectionner des mots-clés de l'UAT ? Pour y parvenir, nous essayons dans cette étude de créer un système de recommandation de $k$ mots-clés pertinents issus de l'UAT, dans le but d'assister les auteurs dans leur choix de mots-clés et faciliter la découverte bibliographique.

## 1.2  État de l'art

L'extraction d'information dans les articles d'astrophysique est au cœur des intérêts de SciX (Science Explorer, précédemment NASA/ADS (Astrophysics Data System)), qui répertorie des millions d'articles scientifiques. Originellement centré sur l'astronomie, mais récemment ouvert à d'autres domaines tels que les sciences de la Terre, la physique ou la biologie, SciX propose un moteur de recherche s'appuyant sur différentes métadonnées telles que les auteurs, les journaux, les dates de publication et, plus récemment, les mots-clés issus du Unified Astronomy Thesaurus (UAT) (Accomazzi et al., 2014). Cet intérêt croissant pour la découvrabilité automatisée d'articles scientifiques a donné lieu à trois éditions du workshop WASP/WIESP (Workshop for Artificial Intelligence for Scientific Publications / Workshop on Information Extraction from Scientific Publications) en 2022 (Ghosal et al., 2022), 2023 (Ghosal et al., 2023) et 2025 (Accomazzi et al., 2025).

Afin de démocratiser l'usage des mots-clés UAT dans le référencement scientifique, SciX a récemment ajouté un champ de métadonnées pour les mots-clés UAT générés par leur modèle **KAILAS** (Keyword Labeler At SciX) [2], modèle que nous évaluons dans cette étude bien qu'il soit basé sur une version antérieure de l'UAT (v5.1.0) [3].

**Classification hiérarchique de textes.** Le problème de classification textuelle multi-labels sur des taxonomies hiérarchiques (HTC, *Hierarchical Text Classification* (Wang et al., 2023)) a fait l'objet de nombreux travaux récents. (Feng et al., 2024) proposent d'enrichir les représentations textuelles avec des connaissances externes. Leur modèle, AMKI-HTC, incorpore deux niveaux de connaissance

---

1. https://journals.aas.org/news/aas-journals-uat/

2. Les caractéristiques de KAILAS et ses données d'entraînement sont décrites ici : https://astrothesaurus.org/wp-content/uploads/2025/12/UATSplinter-KAILAS.pdf.

3. Les différentes versions ainsi que l'historique des modifications sont consultabes sur GitHub : https://github.com/astrothesaurus/UAT/.

(micro et macro) de manière adaptative pour améliorer les performances sur trois benchmarks HTC. On distingue généralement deux approches pour la HTC : l'approche *globale*, qui apprend à prédire simultanément plusieurs nœuds de la hiérarchie avec un seul modèle, et l'approche *locale*, qui construit plusieurs classifieurs spécialisés par niveau hiérarchique. (Jiang et al., 2022) proposent HBGL, un modèle hybride qui exploite à la fois la hiérarchie globale (structure statique contenant tous les labels) et la hiérarchie locale (sous-ensemble de labels pertinents pour chaque document). HBGL s'appuie sur BERT pour encoder les informations sémantiques et hiérarchiques, obtenant des améliorations significatives par rapport à l'état de l'art.

**Représentations vectorielles et réseaux de neurones sur graphes.** L'utilisation de représentations vectorielles pour la HTC a été étudiée par (Stein et al., 2019), qui comparent différentes méthodes de vectorisation (GloVe, word2vec, fastText) combinées à plusieurs algorithmes d'apprentissage. Leurs expériences sur le dataset RCV1 montrent que fastText atteint un score lcaF1 de 0.893. Plus récemment, les Graph Neural Networks (GNN) ont montré leur pertinence pour la classification de textes. (Lin et al., 2024) proposent LaGCN, un modèle basé sur des Graph Convolutional Networks qui intègre des nœuds représentant les classes elles-mêmes dans le graphe, capturant ainsi les corrélations document-mot, mot-mot et mot-classe.

**Lissage de labels.** Le lissage de labels (*label smoothing*) adapté aux graphes constitue une technique prometteuse pour améliorer la généralisation des modèles. En s'inspirant des travaux de (Zhou et al., 2003) sur l'apprentissage, (Wang et al., 2021) proposent une méthode de lissage tenant compte de la structure du graphe, avec cohérence locale et globale. Cette technique de propagation de labels permet de lisser les annotations en diffusant l'information le long des arêtes du graphe, ce que nous avons tenté d'exploiter dans notre approche par une propagation itérative sur le graphe UAT.

**Modèle de langue spécialisé.** Dans le domaine de l'astrophysique, (Grezes et al., 2021) ont développé AstroBERT, un modèle de langue pré-entraîné sur un large corpus d'articles d'astrophysique. Ce modèle capture les spécificités du vocabulaire et des formulations propres au domaine, offrant de meilleures performances qu'un modèle BERT générique pour des tâches d'extraction d'information et de classification en astrophysique. Nous exploitons ce modèle pour vectoriser les nœuds du graphe et les documents à classifier dans plusieurs de nos expériences.

# 2  Données

## 2.1  Prépublications héliophysique

En tant que laboratoire d'héliophysique, notre intérêt se porte sur les articles de ce sous-domaine pour vérifier l'applicabilité des différentes chaînes de traitement à notre cas d'usage. Ainsi, nous avons annoté un ensemble de 31 articles d'héliophysique publiés dans le livre des proceedings de la conférence Planetary Radio Emissions IX (Louis et al., 2023) avec des mots-clés UAT que nous jugeons pertinents[4]. Ces articles sont indexés dans ADS, mais absents de notre jeu de données d'entraînement car ils ne sont pas annotés en UAT sur ADS. Ils ont potentiellement déjà été vus par des modèles tels qu'AstroBERT ou KAILAS, qui ont été entraînés sur davantage d'articles. Un exemple de résumé que nous avons annoté en interne est présenté annexe B.

---

4. Un annotateur, qui est aussi co-auteur de la revue, a sélectionné quelques mots-clés, puis deux personnes ont vérifié leur pertinence

## 2.2 Jeu d'entraînement ADS

### 2.2.1 Corpus astrophysique

Le corpus d'entraînement ayant servi pour les expériences mentionnées dans cet article est disponible sur HuggingFace (Sazuna/UAT_keywords). Il est constitué de 34025 textes (titre + abstract). Chaque article possède 3 à 7 mots-clés en moyenne choisis par les auteurs (colonnes `uat_uri`, `uat_label` et `multihot`), puis a été augmenté par une correspondance de chaîne de caractères entre le texte et les labels des concepts UAT ainsi que leurs synonymes, ajoutant 3 à 9 nouveaux labels en moyenne par article (colonnes `uat_uri_extended`, `uat_label_extended` et `multihot_extended`). L'ajout de ces colonnes a pour but de ne pas "désapprendre" aux différents modèles à faire des correspondances de chaînes de caractères, mais nous ne les avons pas encore exploitées dans cette étude.

Ce corpus a été construit en requêtant l'API de SciX. Pour chaque concept UAT, nous créons une requête filtrant sur les mots-clés avec :
— le label du concept ;
— son URI.
Cela permet de ne récupérer que les articles explicitement annotés en UAT par les auteurs ou éditeurs, assurant une certaine qualité des annotations.

### 2.2.2 Sous-corpus héliophysique

Les tests (inférence) ont été effectués sur le sous-corpus héliophysique, qui contient tous les articles du corpus astrophysique ayant au moins un mot-clé sous le concept "Heliophysics" dans la hiérarchie de l'UAT pour un total de 5361 articles. Comme ils ne contiennent au minimum qu'un mot-clé d'héliophysique, il n'est pas certain que tous ces articles relèvent bien de l'héliophysique, contrairement au corpus de prépublications héliophysique.

## 2.3 Défis

### 2.3.1 Défis liés à l'Unified Astronomy Thesaurus

L'un des défis lié à l'UAT est qu'il évolue au fil des versions : nouvelles définitions, nouveaux mots-clés, restructuration... Par exemple, dans sa version 6, le concept d'héliophysique a été ajouté à la racine, impliquant le déplacement de concepts existants sous cette nouvelle branche. Ainsi, un modèle se basant sur les relations hiérarchiques devra être entièrement reconstruit à chaque nouvelle version.

Le grand nombre de mots-clés constitue un défi de taille, car il implique au modèle d'être capable de gérer une forte diversité lexicale, de distinguer des termes parfois très proches sémantiquement et de maintenir de bonnes performances malgré la complexité croissante de l'espace de classification.

Enfin, l'UAT v6.0.0 est encore sémantiquement pauvre : il possède de brèves définitions techniques sur les concepts les plus importants, aucune définition sur les autres, ce qui diminue son applicabilité à des tâches de traitement automatique des langues. L'UAT relie les concepts entre eux au moyen de deux types de relations : verticales (`skos:broader` et `skos:narrower`) et horizontales

(skos:related), qui s'apparentent souvent à un lien de quasi-synonymie, par exemple entre les concepts "Meteors" (1038) et "Meteorites" (1041).

### 2.3.2 Défis liés aux corpus

Comme indiqué précédemment, nous n'avons récupéré que les articles pour lesquels les auteurs ont explicitement défini des concepts UAT. Cependant, cela résulte en un corpus relativement restreint en taille (seulement 34 025 articles annotés). De plus, une trentaine de mots-clés pourraient être considérés comme acceptables pour un article, sans figurer pas parmi les mots-clés définitifs de cet article.

Un autre corpus similaire de petite taille a été publié par SciX (adsabs/SciX_UAT_keywords). Cependant, KAILAS n'a pas été entraîné sur ces données, qui sont trop peu nombreuses, mais sur un corpus beaucoup plus grand (200 000 articles), construit à partir des mots-clés et des labels de l'UAT sans vérification. Le problème devient alors la qualité des annotations : un article annoté avec "Planets" peut faire référence à plusieurs concepts, celui de planète du système solaire (1260 et celui d'exoplanète (498. De même le mot-clé "Jupiter" peut faire référence à la planète Jupiter 873 ou aux exoplanètes de type joviennes 753, dont les caractéristiques se rapportent à celles de Jupiter (par exemple : "Hot Jupiter"). Ces erreurs d'annotation sont dues au fait que les auteurs ignorent l'existence de ces ambiguïtés dans l'UAT.

Un autre défi lié à notre corpus astrophysique est la mauvaise distribution des labels, comme présenté annexe 2. On peut voir que parmi les 2411 concepts UAT, seuls 1600 environ sont représentés et ce de façon très déséquilibrée. La solution consisterait en une augmentation artificielle de données, ou en l'application d'une méthode de classification non-supervisée.

# 3 Description des expériences

Le code utilisé pour effectuer ces expériences est disponible sur GitHub (Fretel et al., 2026).

## 3.1 Baseline KAILAS

Afin de mesurer l'apport de ces expériences, nous comparons nos résultats avec ceux du modèle KAILAS. N'ayant pas été entraîné sur la dernière version de l'UAT, KAILAS ne renvoie pas de nouveaux labels (2372 dans la v5.1.0 de l'UAT au lieu de 2411 dans la v6.0.0). Dans la configuration par défaut, nous gardons les dix meilleurs labels, mais nous coupons la phrase afin qu'elle respecte la taille limite du contexte imposé par le modèle. Dans la configuration "phrase", pour chaque titre + abstract, KAILAS renvoie un label par phrase (découpage simple sur les points). Le tutoriel KAILAS recommande ce découpage par phrase pour éviter une surcharge car le contexte du modèle est limité à 512 tokens. Nous avons évalué les deux types de découpage.

## 3.2 Classifieur TF-IDF

Pour cette expérience, illustrée figure 1, nous transformons d'abord chaque texte du jeu de données en features en commençant par vectoriser les documents avec un module TF-IDF (term frequency inverse document frequency). Nous limitons la taille du vocabulaire à cent mille n-grammes allant de 1 à 3 mots. En considérant $tf_{i,j}$ la fréquence d'un terme $i$ dans le document courant $j$, $df_i$ la fréquence de documents contenant ce terme dans tout le corpus, et $N$ le nombre total de documents du corpus, la formule du TF-IDF est donnée ci-dessous :

$$tf\text{-}idf_{i,j} = tf_{i,j} \times \log\left(\frac{N}{df_i}\right)$$

Puis, nous utilisons le TruncatedSVD de scikit-learn (décomposition en valeurs singulière tronquée) pour réduire le nombre de paramètres à 8192. Ce module permet de garder les paramètres les plus importants en réduisant drastiquement le nombre de dimensions, accélérant l'apprentissage.

Enfin, nous entraînons un modèle de régression linéaire prédisant un score de probabilité pour chacun des 2411 concepts de l'UAT version 6.0.0. Cependant, nous remarquons que le modèle sur-apprend à partir de la deuxième époque (fonction de perte croissante sur le jeu de validation), malgré nos tentatives d'ajustement des hyper-paramètres comme la pondération par classe (BCEWithLossDigits + pos_weight) ou encore un taux d'apprentissage (learning rate) variable ($1 \times 10^{-4}$, avec comme dégradation des pondérations (weight decay) : $1 \times 10^{-5}$.

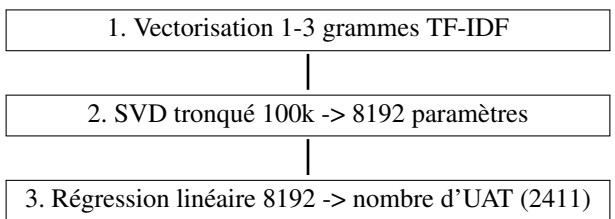

FIGURE 1 – Chaîne de traîtement du classifieur TF-IDF avec régression linéaire.

## 3.3 AstroBERT

Pour cette expérience, nous effectuons un calcul de similarité cosinus entre les articles (abstracts dans notre cas) et les représentations textuelles des concepts UAT. Pour générer les plongements lexicaux, nous utilisons AstroBERT (Grezes et al., 2021), un modèle BERT basé sur RoBERTa ré-entraîné sur plus de 400 000 articles d'astrophysique.

Afin de pouvoir détecter des concepts à l'intérieur de phrases, nous générons une représentation vectorielle par phrase plutôt que de prendre le résumé dans sa globalité et de récupérer les 10 labels les plus probables. Comme il manque beaucoup de définitions dans l'UAT, nous avons remarqué que ce modèle renvoyait toujours les mêmes concepts et en avons conclu qu'AstroBERT ne vectorise pas chaque concept de façon équitable. Pour y remédier, nous avons demandé à un LLM : `phi4:14B`

([Abdin et al., 2024](#)) de générer une définition pour chaque concept non défini, puis nous avons re-vectorisé cette nouvelle version de l'UAT avec AstroBERT. Les résultats des deux expériences sont présentées table 1. Ce modèle a été retenu en raison de son bon compromis entre taille et qualité de génération sur des tâches de raisonnement scientifique.

## 3.4   R-GCN

Pour cette expérience, nous avons entraîné un R-GCN (Relational Graph Convolutional Network, défini par ([Schlichtkrull et al., 2017](#)), dont l'architecture est identique au graphe de l'UAT. Chaque nœud du graphe $n$ correspond à un concept UAT, chaque arête $e$ correspond à une relation dans la matrice d'adjacence $A_{i,j}$ du graphe de l'UAT et chaque type d'arête correspond à un type de relation parmi `SKOS:broader`, `SKOS:narrower` et `SKOS:related`, qui sont les trois types de relations reliant les concepts UAT. Les vecteurs d'origine des nœuds sont calculés en vectorisant le label et la définition des concepts avec AstroBERT, de même que les documents en entrée du modèle.

L'avantage théorique d'un graphe neuronal convolutionnel (GCN) est qu'il est capable de propager des informations récursivement en suivant les relations du graphe : à chaque couche de convolution (généralement limitée à une ou deux pour éviter le sur-lissage), les matrices des nœuds sont mises à jour en effectuant une opération, par exemple une somme des matrices voisines. En R-GCN, le principe est le même, mais il existe autant de matrice de convolution qu'il n'y a de type de relation (trois dans notre thésaurus), ainsi un nœud relié à deux nœuds "enfants" et un nœud "parent" verra ses poids mis à jour de deux façons différentes.

## 3.5   Correspondance de chaîne de caractères

Les méthodes citées ci-dessus se basent uniquement sur des vectorisations et des apprentissages supervisés, les rendant très dépendants de la qualité et de la diversité des données d'entraînement. Afin de détecter les concepts peu fréquents, nous avons testé une simple extraction de mots-clés par correspondance de chaîne de caractères. Cette correspondance s'effectue en priorité sur le label principal, puis sur les labels alternatifs. Elle détecte en priorité le label correspondant le plus long, par exemple une mention de "slow solar wind" ne sera non pas associée au concept "Solar wind" (1534), mais bien à son concept fils "Slow solar wind" (1873). Une détection de mots-clés en astronomie par correspondance de chaînes de caractères est aussi implémentée sur [OntoPortal-Astro](#) ([Cecconi et al., 2025](#)) (ouvrir l'onglet "Annotator", puis sélectionner l'ontologie UAT).

# 4   Résultats

## 4.1   Métriques

Pour tous les résultats, la F-mesure, la précision et le rappel sont agrégés en micro moyennes, c'est-à-dire en moyennant sur chaque classe individuellement. Pour le classifieur TF-IDF, R-GCN et AstroBERT, les résultats sont mesurés sur les top-k meilleurs labels (avec $k = 10$) plutôt que sur des labels dépassant un seuil minimum, car les scores sont très variables d'un texte à l'autre.

Le nombre de labels en sortie varie pour les trois expériences suivantes : label match renvoie tous les labels présents dans le texte. KAILAS baseline (phrases) et AstroBERT (phrases + def) renvoient un label par phrase. Enfin, "def" (AstroBERT) signifie que l'on a utilisé un LLM (phi4:14b (Abdin et al., 2024)) pour générer une définition des concepts UAT avant leur vectorisation.

## 4.2 Résultats

| Résultats (moyenne micro) | ADS sous-corpus héliophysique | | | Prépublication héliophysique | | |
|---|---|---|---|---|---|---|
| | Précision | Rappel | F1 | Précision | Rappel | F1 |
| KAILAS baseline | 0.3942 | 0.8841 | **0.5453** | 0.3000 | 0.4537 | 0.3612 |
| KAILAS baseline (phrases) | 0.3561 | 0.4582 | 0.4008 | 0.3602 | 0.3268 | 0.3427 |
| Classifieur TF-IDF | 0.3619 | 0.8116 | 0.5006 | 0.5586 | 0.8100 | **0.6612** |
| Astrobert (phrases) | 0.0027 | 0.0053 | 0.0035 | 0.0041 | 0.0049 | 0.0045 |
| AstroBERT (phrases + def) | 0.0015 | 0.0027 | 0.0020 | 0.0044 | 0.0049 | 0.0046 |
| R-GCN | 0.0020 | 0.0044 | 0.0027 | 0.0097 | 0.0146 | 0.0117 |
| Label match | 0.2210 | 0.2674 | 0.2420 | 0.3136 | 0.2585 | 0.2834 |

TABLE 1 – Comparaison des résultats sur nos différentes expériences

### 4.2.1 Baseline KAILAS

Dans la configuration "phrases" (un mot-clé par phrase), KAILAS renvoie certains mots-clés sans rapport avec l'article. Par exemple, pour l'article avec le DOI 10.25546/104047 intitulé « Auroral emissions and inner magnetospheric dynamics during Earth's response to the 28th October 2021 Coronal Mass Ejection », KAILAS renvoie des mots-clés comme "Neutron stars" (1108), alors que l'article fait référence à la couronne du Soleil et non pas à une étoile à neutron.

Dans la configuration par défaut, c'est-à-dire en renvoyant les dix premiers labels, KAILAS performe beaucoup mieux sur le sous-corpus ADS héliophysique, passant d'un score F1 de 0.4008 à 0.5453. Nous pensons qu'il arrive mieux à capter le contexte des résumés dans leur entièreté (avec un troncage à 512 tokens). Un exemple de labels obtenus en sortie de KAILAS est présenté annexe C.

### 4.2.2 Classifieur TF-IDF

Notre classifieur TF-IDF avec TruncatedSVD et régression linéaire a surpassé les performances de KAILAS sur le corpus de prépublication, mais est légèrement en dessous sur le sous-corpus ADS héliophysique. En comparant les sorties des deux modèles, nous avons remarqué que sur nos deux corpus d'héliophysique, KAILAS sort parfois des concepts liés à l'interstellaire (sans relation avec l'héliophysique) tandis que les labels négatifs renvoyés par le classifieur TF-IDF restent acceptables. Par exemple, dans notre corpus de validation (prépublication), les deux modèles assignent uniquement des labels pertinents pour l'article ayant pour DOI 10.25546/103093 : « Plasma waves in the very local interstellar medium : a brief review ». Un exemple de labels obtenus est présenté annexe D.

### 4.2.3 AstroBERT

La vectorisation avec AstroBERT avec une mesure de similarité cosinus n'a nullement résolu le problème, comme le démontrent les résultats. AstroBERT est très sensible à la qualité des définitions des concepts UAT. En conséquence, AstroBERT renvoie presque toujours les mêmes mots-clés pour chaque article.

### 4.2.4 R-GCN

Pour cette expérience, l'amélioration que nous avions anticipée n'a pas été observée. En effet, nous avions émis l'hypothèse qu'en apprenant à renvoyer des nœuds ensemble, le modèle apprendrait des corrélations entre les nœuds proches, mais un problème plus important a joué en notre défaveur : le manque crucial de données d'entraînement, entraînant un sous-apprentissage et une stagnation rapide de la courbe d'entraînement, ainsi que le renvoi systématique des mêmes labels à chaque appel.

Pour pallier ce problème de sous-apprentissage, une approche de lissage de labels (Zhou et al., 2003) basée sur la structure du graphe de l'UAT (similairement à (Wang et al., 2021)) a été testée afin de moins sanctionner le modèle lorsqu'un nœud voisin d'un nœud positif est renvoyé : soit $Y^{(t)} \in \mathbb{R}^{n \times k}$ la matrice des labels (multi-hot) à l'itération $t$, et $A \in \mathbb{R}^{n \times n}$ une matrice d'adjacence. À chaque itération, les labels sont propagés via une multiplication matricielle $Y^{(t)}A$, qui diffuse l'information vers les voisins dans le graphe, puis combinés avec les labels actuels via une interpolation convexe contrôlée par un paramètre $\alpha \in [0, 1]$ :

$$Y^{(t+1)} = \alpha Y^{(t)} + (1 - \alpha)Y^{(t)}A$$

Tout comme pour le nombre de couches du modèle convolutif, il est préférable de se limiter à une ou deux itérations pour éviter le sur-lissage des labels. Cette méthode de lissage est à expérimenter conjointement avec d'autres architectures d'apprentissage (régression linéaire, etc) afin de voir si l'on observerait une amélioration des performances ou une convergence plus rapide des modèles vers leurs poids optimaux.

Par la suite, nous avons ajouté une relation self-self allant de chaque nœud à lui-même afin que les nœuds tiennent compte de leur état précédent, puis nous avons ajouté un nœud abstrait connecté à tous les autres nœuds du graphe par une relation asymétrique `node2global` et `global2node`. La matrice vectorielle de ce nœud représente la sémantique du graphe entier.

Cependant, aucune de ces modifications n'a eu d'impact sur l'apprentissage, nous avons donc fait l'hypothèse que le problème provenait des données (taille du jeu d'entraînement limité, annotations peu qualitatives, non équilibrées avec une grande partie des classes n'ayant aucun exemple).

### 4.2.5 Correspondance de chaîne de caractères

Cette méthode est, selon nous, complémentaire aux autres car elle pourra permettre la détection des concepts rares que les autres modèles n'ont pas réussi à détecter puisqu'ils sont par définition probabilistes. Les mots-clés détectés, bien que peu nombreux, sont quasiment certains d'être pertinents (voir résultats annexe E). Nous avons toutefois observé certains faux positifs récurrents, comme "A stars" (5), lié au fait que nous avons autorisé une correspondance avec des labels au singulier et ignoré la casse.

# 5   Conclusion

Dans cette étude, nous avons constitué un corpus d'articles d'astrophysique. Nous avons comparé plusieurs méthodes de classification multi-labels en les appliquant à un sous-corpus d'héliophysique afin d'évaluer leur capacité de spécialisation.

À l'analyse des résultats, il apparaît que la classification multi-labels de résumés d'articles en astrophysique (et en héliophysique) constitue une tâche difficile. Elle requiert des données d'apprentissage à la fois abondantes et de bonne qualité, notamment en limitant les ambiguïtés d'annotation liées au caractère interdisciplinaire de l'astrophysique et en enrichissant l'annotation de tous les labels pertinents, non pas limités aux seuls labels proposés par les revues ou les auteurs. Les scores globalement faibles observés pour la plupart des méthodes (AstroBERT, R-GCN) semblent principalement dûes à ce manque. Dans d'autrs domaines, et avec un nombre de labels comparables, les scores de micro-F1 dépassent rarement 0.7, comme dans cette étude sur le domaine de la législation européenne (Chalkidis et al., 2019) (7000 concepts).

Parmi les approches testées, la méthode fondée sur TF-IDF associée à un classifieur linéaire a donné les résultats les plus prometteurs. À l'inverse, les performances les plus faibles ont été obtenues avec la mesure de similarité cosinus appliquée aux plongements lexicaux d'AstroBERT, ainsi qu'avec l'entraînement d'un R-GCN dans lequel chaque nœud du graphe représente un concept UAT. Lors de l'entraînement de ce modèle, un phénomène de sous-apprentissage a été observé.

## Perspectives

Dans nos futurs travaux, nous aimerions collecter un plus grand corpus et faire de la classification sur les articles entiers, plutôt que seulement sur le titre et le résumé. Pour améliorer le score des classes sous-représentées, nous voudrions expérimenter une augmentation de données par des LLMs, ou mettre en place un recommandeur hybride proposant des concepts issus à la fois de la correspondance des chaînes de caractères et d'un classifieur sémantique tel que KAILAS. Enfin, nous continuerons d'explorer les réseaux neuronaux sous forme de graphe, en s'inspirant notamment de (Lin et al., 2024) à partir de PMI (Pointwise Mutual Information) et TF-IDF entre documents et labels. Lors de notre expérience sur TF-IDF, une valeur N-PMI (Normalized Pointwise Mutual Information) a été ajoutée pour mesurer la cohérence entre les dix labels renvoyés par le classifieur (voir en bas de l'annexe D). Cette valeur est mesurée à partir des décomptes de co-occurrences entre les labels dans le jeu d'entraînement et le décompte d'occurrence total d'un label. Ainsi, si un label n'apparaît qu'une seule fois dans tout le corpus et qu'il ne co-occure qu'avec trois autres labels, il aura une valeur N-PMI élevée avec ces trois labels. Dans ce cas, serait pertinent de l'ajouter aux mots-clés candidats.

# Remerciements

Ce travail a bénéficié du soutien financier du programme OPAL, financé par le projet OSCARS (Convention de subvention : 101129751). Les auteurs remercient également le CNRS et l'Observatoire de Paris pour leur soutien. De plus, les auteurs tiennent à remercier Felix Grezes (SciX) pour ses éclaircissements sur ses travaux menés sur KAILAS et AstroBERT.

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

# A    Distribution des annotations dans le corpus

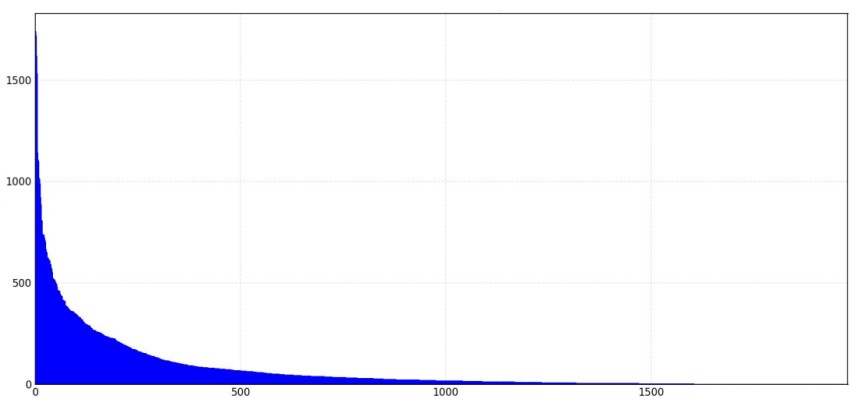

FIGURE 2 – Distribution des mots-clés UAT de notre corpus collecté à partir d'ADS. Axe X : mots-clés, axe Y : nombre d'occurrences.

# B  Exemple d'article annoté du corpus héliophysique prépublié

DOI : 10.25546/103093
Titre : « Plasma waves in the very local **interstellar medium** : a brief review. »

Résumé : "The Plasma Wave Science instruments on the two Voyager spacecraft are providing observations of plasma waves in the very local **interstellar medium**. One of the most important aspects of these observations are the inferred electron densities from the identification of electron plasma oscillations and a quasi-thermal line at the plasma frequency. They provide the first and only in situ determination of the plasma density in the interstellar medium and have revealed an outward radial gradient as a large scale feature of the interstellar medium upstream of the **heliopause**. The presence of the electron plasma oscillations provides evidence that transients from the **sun**, after propagating through the **heliosphere**, can modulate the surrounding medium."

Concepts annotés :
— 851 (Interstellar plasma)
— 847 (Interstellar medium)
— 1544 (Space plasmas)
— 848 (Interstellar medium wind)
— 707 (Heliopause)
— 711 (Heliosphere)
— 2089 (Plasma physics)

# C  Sortie du classifieur KAILAS (top 10) sur cet exemple

Concepts prédits :
— 106 (Astrosphere interstellar medium interactions)
— 1534 (Solar wind)
— 1544 (Space plasmas)
— 1690 (Termination shock)
— 707 (Heliopause)
— 710 (Heliosheath)
— 711 (Heliosphere)
— 845 (Interstellar magnetic fields)
— 847 (Interstellar medium)
— 851 (Interstellar plasma)

# D  Sortie du classifieur TF-IDF + régression linéaire (top 10) sur cet exemple

Concepts prédits :
1. 0.5936 : 851 (Interstellar plasma)
2. 0.5873 : 106 (Astrosphere interstellar medium interactions)
3. 0.5741 : 707 (Heliopause)

4. 0.5552 : 848 (Interstellar medium wind)
5. 0.5516 : 1544 (Space plasmas)
6. 0.5427 : 711 (Heliosphere)
7. 0.5398 : 847 (Interstellar medium)
8. 0.5370 : 2089 (Plasma physics)
9. 0.5345 : 710 (Heliosheath)
10. 0.5343 : 1261 (Plasma astrophysics)

N-PMI moyen des top 10 concepts : 0.3321

# E   Sortie de la correspondance de chaînes de caractères sur cet exemple

Concepts détectés :
— 847 (Interstellar medium)
— 707 (Heliopause)
— 1693 (the Sun)
— 711 (Heliosphere)

Notons que si le terme "Interstellar" avait été ajouté dans le titre devant "Plasma", le concept "Interstellar plasma" (qui fait partie des mots-clés que l'on veut voir apparaître C aurait pu être détecté ici. Cela démontre la nécessité de l'application de méthodes sémantiques plus poussées.