# OpenReview forum: "Evaluation of Methods for Assigning Standardized Astronomy Keywords to Heliophysics Papers' Abstracts"
_ls2n.fr/CORIA-TALN/2026/Workshop/ARTS — ls2n CORIATALN 2026 Workshop ARTS Submission_

### Official Review · Reviewer_RrJK · 2026-04-24

**Mode De Presentation:** Poster

**Confience:**

Oui

**Decision:**

Accepté

**Relecture:**

*Information: Cet article n'était pas correctement anonymisé pendant la phase de relecture, ceci n'a pas eu d'incidence sur mon travail de relecture et n'a pas influencé mon avis.*

J’ai trouvé l’article globalement clair dans sa présentation, ce qui est appréciable étant donné qu’il traite d’un domaine très spécialisé que je ne maîtrise pas. La contribution me semble pertinente compte tenu de la difficulté de la tâche et l'analyse des modèles existant semble raisonnable.

Néanmoins, certains aspect du papier pourraient être améliorés :

- Les informations relatives aux problématiques liées à l’UAT sont trop dispersées dans le document, ce qui complique l’évaluation rapide de la difficulté de la tâche et nuit à la compréhension globale (tailles de labels dans l'intro, description dans le SOTA, présentation des liens de relations dans la section R-GCN, + problématiques liées au versioning, labels alternatifs, etc.).
Il serait pertinent de faire une section dédiée à la description complète et détaillée de l'UAT présentant toutes les problématiques associées puis de revenir sur chacune d'entre elles par la suite.

- Des définitions d'étiquettes ont été générées via  phi4-14B. Deux points ne sont pas abordés dans cet article, premièrement le choix du modèle mériterait d'être explicité, pourquoi celui-ci en particulier?
Également, un protocole évaluatoire devrait être réalisé pour assurer la qualité des définitions. Il me semble important de valider un sous-ensemble de définition avec des experts du domaines afin de mesurer à quel points les modèles LLMs sont en mesure de réaliser cette tâche.

- Concernant le jeu de données annoté, le nombre, profil des annotateurs ainsi que le protocole d'annotation devrait être mis à disposition pour permettre la réplicabilité de l'annotation mais également de mieux comprendre pourquoi d'aussi grandes différences entre annotations et labels auteurs. Sans remettre en question l'effort réalisé par les annotateurs, il est difficle de mesurer la fiabilité de ce travail d'annotation sans ces informations.
Compte tenu de la complexité de la taxonomie, il est possible que les auteurs comme les annotateurs ne savent pas bien annoter les documents, mieux ancrer l'annotation permettrait d'avoir une meilleure vision sur les différences notables entre auteurs et annotateurs.

- Étant donné l’ampleur de la taxonomie, je ne sais pas si on peut affirmer qu’un score de F1 supérieur à 50 % soit réellement faible. Dans le domaine des articles en TAL, des tâches similaires reposant sur des taxonomies plus restreintes obtiennent des scores certe plus élevés, mais qui s’explique probablement par un nombre de label beaucoup plus réduit. (ex. https://aclanthology.org/2025.acl-long.1224.pdf).
Autre exemple, une de mes expérimentations (article en cours de revue à TALN donc pas disponible pour le moment) sur la typologie de l'ACL Rolling Review permettant de marquer les contributions réalisées par les auteurs d'un article présente des résultats de ~60% pour TF-IDF et de ~74% de F1 micro pour les meilleurs modèles (pour 11 étiquettes / multi-étiquette). Expliquer les scores en faisant référence à d'autres études travaillant sur une tâche proche permettrait de mettre en pespective ces résultats.

- Une analyse des erreur un plus poussées des meilleurs modèles pourrait permettre une meilleure compréhension des zones d'ombres liées à la taxonomie. Les erreurs sont-elles "proches" et liées à des problèmes de ségmentation dans l'abstract (ex: A star vs B dwarf star) ou au contraire des concepts très éloignés?
Compte tenu de la taille conséquente de la taxonomie, le fait de guider les auteurs vers des labels "normalisés" qu'ils n'auraient pas envisagés pourrait également être intéressant

- Plusieurs coquilles : "Cette étude s’intéresse l’attribution", "L’avantage théorique d’un en graphe", "les résulats sont mesurés", "notre sous-coprus", "résentés annexe". Faire une relecture attentive de l'article.

**Resume:**

Cet article aborde la problématique de la prédiction multi-étiquette éparse sur l'*Unified Astronomy Thesaurus*. La problématique principale est la taille de cette taxonomie (2411 étiquettes) et un manque de données lié à cette tâche.
Les auteurs proposent un nouveau corpus d'évaluation permettant de mesurer les performances de différents systèmes d'attribution de mots-clés.

---

### Official Review · Reviewer_GgLW · 2026-05-04

**Mode De Presentation:** Poster

**Confience:**

Oui

**Decision:**

Accepté

**Relecture:**

Cette soumission propose différentes approches pour associer les mots-clés aux articles d'astrophysique. La motivation du travail est lié à un besoin existant dans le domaine pour indexer les articles avec le thésaurus UAT et avoir ainsi un accès plus direct aux publications selon leurs thématiques.

La publication a le mérite de tester plusieurs approches pour l'indexation contrôlée (avec un thésaurus) des articles.
Il est intéressant de voir que l'approche standard en RI et indexation (TF-IDF) reste performante dans ce paysage. De même, le modèle KAILAS, entraîné sur une version antérieure d'UAT, fournit également des résultats compétitifs.

La présentation du travail gagnerait en clareté si le rôle de chaque corpus était décrit avec plus de clareté.

Les auteurs indiquent que certaines approches sont plutôt complémentaires aux autres. Ceci étant dit, est-ce que les auteurs ont essayé de combiner ces différentes approches enter elles ? Si oui, de quelles manières ?


L’avantage théorique d’un en graphe neuronal convolutionnel ->
L’avantage théorique d’un graphe neuronal convolutionnel

**Resume:**

Les auteurs s'intéressent à l'indexation automatique de publications scientifiques avec les mots-clés du Unified Astronomy Thesaurus (UAT). Plus particulièrement, 5 approches sont testées.

Les auteurs travaillent avec la littérature scientifique du domaine. Plusieurs jeux de données ont été constitués :
1. un corpus d'entraînement (34025 titres + résumés), qui est disponible sur HuggingFace. Ce corpus contient les mots-clés UAT définis par les auteurs ou les éditeurs ;
2. un corpus de test avec des articles liés à l'héliophysique (5361 titres + résumés), également avec les mots-clés UAT ;
3. 31 articles de la conférence Planetary Radio Emissions IX annotés manuellement avec les mots-clés de UAT. Ce corpus semble être utilisé pour le test et l'évaluation.

5 approches pour l'association de mots-clés aux articles sont exploitées :
- baseline : le modèle KAILAS (Keyword Labeler At SciX), entraîné sur une version antérieure d'UAT,
- classifieur TF-IDF,
- AstroBERT, avec la génération des définitions pour les termes UAT qui manquaient dans le thésaurus,
- R-GCN (Relational Graph Convolutional Network) entraîné sur UAT,
- correspondance de chaînes de caractères.

Les résultats montrent que le classifieur TF-IDF et KAILAS obtiennent les meilleurs résultats.
Les auteurs considèrent que le faible volume de données est la limite principale du travail actuel.

Le code utilisé pour l'association de mots-clés UAT se trouve sur GitHub.

---

### Decision · Program_Chairs · 2026-05-07

Accept (Poster)